# Growth Quality and Development of Olive Plants Cultured In-Vitro under Different Illumination Regimes

**DOI:** 10.3390/plants10102214

**Published:** 2021-10-18

**Authors:** Pablo Díaz-Rueda, Manuel Cantos-Barragán, José Manuel Colmenero-Flores

**Affiliations:** Instituto de Recursos Naturales y Agrobiología, Spanish National Research Council (CSIC), Av. Reina Mercedes 10, 41012 Sevilla, Spain; cantos@irnase.csic.es

**Keywords:** LED illumination, fluorescent illumination, *Olea europaea*, micropropagation, internodal elongation

## Abstract

Light-emitting diodes (LEDs) are useful for the in-vitro micropropagation of plants, but little information is available on woody species. This work compares the effects of light quality and intensity on the growth and development of micropropagated olive plants from two different subspecies. Illumination was provided with fluorescent and LED lamps covering different red/blue ratios (90/10, 80/20, 70/30, 60/40) or red/blue/white combinations, as well as different light intensities (30, 34, 40, 52, 56, 84, 98 and 137 µmol m^−2^ s^−1^ of photosynthetic photon fluxes, PPF). Olive plants exhibited high sensitivity to light quality and intensity. Higher red/blue ratios or lower light intensities stimulated plant growth and biomass mainly as a consequence of a higher internodal elongation rate, not affecting either the total number of nodes or shoots. In comparison to fluorescent illumination, LED lighting improved leaf area and biomass, which additionally was positively correlated with light intensity. Stomatal frequency was positively, and pigments content negatively, correlated with light intensity, while no clear correlation was observed with light quality. In comparison with fluorescent lamps, LED illumination (particularly the 70/30 red/blue ratio with 34 µmol m^−2^ s^−1^ PPF intensity) allowed optimal manipulation and improved the quality of in-vitro micropropagated olive plants.

## 1. Introduction

Olive (*Olea europaea*, L.) is a woody crop belonging to the *Oleaceae* family. It is a medium-sized tree of about 4 to 8 m high, depending on the variety. The *Olea europaea* species is found in the Mediterranean area [1] and is the only one of the genus that produces edible fruit [2]. Olive is a crop that is well adapted to Mediterranean drylands, with very acceptable productions and is capable of surviving periods of intense water deficit. It does not tolerate temperatures below −10 °C for prolonged periods of time. It is a plant highly dependent on light, so that light deficiency reduces the number of flowers or their viability. Despite its good climatic adaptation, it is a slow-growing tree, requiring optimal multiplication procedures. Micropropagation has arisen as a successful technique for plant propagation through in-vitro tissue culture technology. An increase of in-vitro culture efficiency for olive requires the optimization of temperature, lighting and ventilation parameters. Of them, light quality and quantity play essential roles in plant photosynthesis, development and organogenesis [3,4,5,6].

Plants respond to light quantity and quality through photoreceptors and photomorphogenic responses [7,8], regulating different aspects of plant development, including seed germination, seedling elongation, vegetative growth and architecture, flowering and senescence [9]. Among photoreceptors, phytochromes absorb light wavelengths from the whole spectrum, but most importantly in the red and far-red regions [10,11]. Phytochromes affect the regulation of phytohormones such as auxin [12] salicylic and jasmonic acid [13,14]. Other plant photoreceptors include cryptochromes and phototropins, which respond to blue and ultraviolet (UV) light. Cryptochromes perceive blue and UV-A light (370–450 nm), regulating many physiological and developmental processes, such as chlorophyll biosynthesis, response to high-irradiance stress and photomorphogenesis [15]. Phototropins mainly mediate phototropic responses triggered by blue and UV-A light in the 320–500 nm wavelength range [16].

Fluorescent lamps, traditionally used for in-vitro culture, increase the temperature of the plant culture chamber, making necessary more expensive cooling systems and therefore higher energy consumption. In contrast, light-emitting diodes (LED) lighting has clear benefits for optimizing the performance of plant growth chambers, including smaller size, longer life, lower temperature and the option to regulate single wavelength emission [17,18,19]. It is well known that the peak spectral output of red and blue LED chips coincide closely with the main absorption peaks of chlorophyll and the reported wavelengths for maximum photosynthetic efficiency [20,21]. Therefore, the use of LED illumination has become considerably more attractive for plant in-vitro culture and, particularly, for micropropagation. Thus, several plant species have been grown successfully in-vitro under LED illumination [22]. They include plantlets of banana [6], cotton [19], *Lippia filifolia* [23] and rapeseed [24], cultured under different light qualities including fluorescent lamps and different combinations of blue vs. red LEDs. In addition, the use of LED illumination for the in-vitro culture of some woody plant species, such as *Eucalyptus urophylla* [25], *Cedrela fissilis* [26], *Pinus sylvestris* and *Abeis borisii-regis* [27], has been reported. These results confirmed positive effects of LED illumination on physiological and morphological parameters of in-vitro grown plants. However, responses vary according to plant species and it is necessary to empirically determine the effects of light quality and intensity resulting from different spectral combinations of LED lighting. 

LED illumination facilitates the use of different light spectra to induce different developmental responses, for example, shoot multiplication, internode elongation, rooting, hardening, or even flowering in different plant species [28,29,30,31,32,33,34,35,36]. Many authors have reported different techniques for in-vitro olive culture, such as somatic embryogenesis [37,38,39], zygotic embryogenesis [40], callus culture [41,42] and micropropagation [43]. Micropropagation for the conservation of olive genetic diversity has been widely applied, and many olive cultivars have been micropropagated successfully. This process is influenced by many factors, mainly, the medium composition, the plant genetic background, the physiological and sanitary state of the mother plant and physical factors, among which light and temperature of the growth chamber are particularly relevant. Related to light influence, scarce information is available on in-vitro culture of olive, particularly light studies have been reported for the improvement of ex-vitro rooting from microplantlets [44]; callus culture [42,45] or the reduction of vitrification [46]. However, no information is available on the effect of light quality and quantity on the in-vitro growth of micropropagated *Olea europaea* plants, nor on the specific benefits of LED lighting. The aim of this study was to evaluate to what extent light quality and intensity regulate developmental, morphological, anatomical, physiological and photosynthetic parameters of micropropagated olive plants. For this purpose, two wild olive genotypes belonging to two different subspecies of *Olea europaea* were micropropagated under different illumination regimes supplied by both fluorescent and LED lighting, and the following parameters were measured: total stem length, internodal length, number of nodes, number of shoots, shoot and callus biomass, total leaf area, number of leaves, stomatal size, stomatal frequency and content of photosynthetic pigments. 

## 2. Results

### 2.1. Effect of Light Quality and Intensity on Stem Growth and Development

A clear variation of plant growth was observed in micropropagated olive plants treated with different light regimes (Figure 1). 

Negative and statistically significant correlations (*p* ≤ 0.001) were observed when light intensity was compared with the total stem length (Figure 2A) and the internodal length (Figure 2C) of AMK34 and GUA7 plants. A higher stem length correlated not only with a lower light intensity, but also with a higher red:blue ratio (Figure 2B,D) in both wild olive subspecies. Thus, plants subjected to greater light intensities or lower red:blue ratio exhibited shorter stems (Figure 2A,B). The effect of light quality and intensity on shoot height was clearly a consequence of the internode length in both genotypes (Figure 2C,D), not affecting either the total number of shoots or nodes per explant (Appendix A). 

Therefore, light quality and light intensity determined independent photomorphogenic responses that could be distinguished from each other, comparing the different combinations of lighting regimes applied. On the one hand, for the same light quality (R:B, 70%:30%), the higher light intensity in the R7B3-52 treatment (52 μmol m^−2^s^−1^ PPF) induced significantly shorter stems (Figure 2B) and shorter internodes (Figure 2D) than the 34 μmol m^−2^s^−1^ PPF light intensity (R7B3-34 treatment) in both olive genotypes. A similar response was observed when the effect of the R6B4W-98 treatment on total stem length (Figure 2B) and internode length (Figure 2D) was compared with that of the R6B4-40 treatment. On the other hand, when similar light intensities (56 and 52 μmol m^−2^s^−1^ PPF) were compared in the R9B1-56 and R7B3-52 treatments, respectively, the 90% red light present in the R9B1-56 treatment determined significantly longer stems (Figure 2B) and internodes (Figure 2D) than the 70% red light content of the R7B3-52 treatment.

Plants subjected to daylight white fluorescent illumination (137 μmol m^−2^s^−1^ PPF) resulted in the shortest internodes and the lowest stem height (Figure 1 and Figure 2) throughout the whole period of plant growth (Appendix A). The strong stem shortening caused by fluorescent lights is a general response in the olive species as could be observed not only in AMK34 and GUA7, but also in other wild olive varieties from the *europaea* subspecies (Appendix A). 

This response determined significant differences in the shoot biomass of AMK34 (Figure 3A,B) and GUA7 (Figure 3D,E) genotypes. Thus, a negative correlation (*p* ≤ 0.001) between the light intensity (PPF) and the dry biomass was observed in in-vitro grown plants (Figure 3A,D). Interestingly, no differences were observed in the callus biomass (Figure 3C,F).

Therefore, it can be concluded that micropropagated olive plants are highly sensitive to light quality and intensity, which regulates the internodal elongation rate, but not the total number of nodes, affecting plant growth and biomass.

### 2.2. Effect of Light Quality on Leaf Growth, Development and Pigment Composition

LED light intensity showed significant positive correlation (*p* ≤ 0.001) with total leaf area (Figure 4A) and leaf fresh weight (Figure 4B) in both AMK34 and GUA7 genotypes. However, the fluorescent light treatment did not fit this correlation, showing abnormally low values of total leaf area and biomass that did not correlate with the equivalent light intensity in the group of plants treated with LED illumination (Figure 4A,B). Contrary to what was observed for the elongation of internodes and stem, light quality did not clearly affect either the total leaf area (Figure 4C,D) or the total number of leaves (Figure 4E,F) in any olive genotype. Thus, different red vs. blue light ratios in R9B1-56 and R7B3-52 treatments having similar light intensities (56 and 52 μmol m^−2^s^−1^ PPF respectively), did not determine differences in either leaf area (Figure 4C,D) or total number of leaves (Figure 4E,F). 

Besides total leaf area, stomatal size and density can affect water relations in plants, which is important when considering the ex-vitro acclimatization success of micropropagated plants. Slight but significant correlations between light intensity and stomatal development could be observed: a negative correlation with the stomatal size (Figure 5A); and a positive correlation with the stomatal frequency (Figure 5B). Thus, a higher light intensity determined a higher density of smaller stomata. Interestingly, the fluorescent light fitted this correlation together with other LED lights, in contrast to the leaf area and biomass responses (Figure 4). Light quality did not clearly affect either the stomatal size or stomatal frequency of the olive variety AMK34 (Figure 5C,D). 

Finally, light intensity showed negative correlations with the content of the photosynthetic pigments chlorophyll (Figure 6A) and carotenoids (Figure 6B) in the AMK34 olive genotype. Light quality affected pigment content since different red:blue ratios in the treatments R9B1-56 and R7B3-52, having similar light intensities, determined significant differences in chlorophyll-A (Figure 6E). Thus, a lower red:blue ratio determined a reduction of chlorophyll-A. Similar trends were observed in chlorophyll-B (Figure 6F) and carotenoids (Figure 6D), although differences were not statistically significant. Thus, lower light intensity and/or a higher red:blue ratio determined a higher content of photosynthetic pigments.

In conclusion, light quality and quantity affected important morphological, anatomical and biochemical parameters, allowing optimal regulation of plant quality during in-vitro olive micropropagation.

## 3. Discussion

*Olea europaea* is widely used worldwide for the production of oil and table olives, so that improving plant propagation procedures is of great interest for the olive industry. Lighting is one of the most important environmental factors determining optimal plant propagation. Knowledge about plant photomorphogenesis has increased drastically due to the growing use of LED illumination [8]. In this work, we show that LED illumination improved the in-vitro growth of olive explants in comparison with fluorescent lamps (Figure 1, Figure 2 and Figure 3). Among different LED lighting regimes, the 70:30 red:blue ratio with 34 μmol m^−2^s^−1^ PPF combined low energy consumption with adequate development in terms of plant growth (Figure 1 and Figure 2), internodal length (Figure 2), biomass (Figure 3) and accumulation of photosynthetic pigments (Figure 6). Adequate internodal length allows optimal manipulation of in-vitro olive explants and therefore higher micropropagation rates. In addition, the R7B3-34 lighting regime reduced the leaf area and biomass (Figure 4), as well as the stomatal density (Figure 5), which is expected to reduce water loss during the sensitive ex-vitro acclimatization process.

The internodal elongation rate of micropropagated olive plants, but not the total number of nodes, was highly sensitive to both light quality and intensity (Figure 1 and Figure 2). Thus, low illumination intensities (e.g., 30–50 μmol m^−2^s^−1^) and/or high red:blue ratios (e.g., 90:10–70:30) enhanced internodal elongation, facilitating the propagation of uninodal explants. Different problems have been associated with the propagation of olive shoots containing short internodal segments: the easy chance to damage axillar meristems, the contact of leaves with the medium and the difficulty to adequately embed the explant into the agar medium. These problems frequently cause necrosis and give rise to undifferentiated structures in micropropagated olive explants. This undesirable growth is induced by high light intensity or low B:R ratios (e.g., R6B4W-98 and, especially, the fluorescent TFL-137 lighting). A negative effect of fluorescent lighting on shoot growth in comparison with LED illumination has also been reported in other plant species [6,21,24,26,47,48]. We show in this work the reduction of shoot biomass (Figure 3A–D) and lower pigments content (Figure 6) in response to higher light intensity, indicating that PPF values higher than 50 μmol m^−2^s^−1^ impairs in-vitro culture of olive plants. Previous reports also observed a decrease of dry weight in plants grown in-vitro at high light intensity [49,50]. This phenomenon is probably the consequence of a suboptimal cell elongation rate under high light intensity, affecting different photosynthetic organs: the internodes, producing abnormally short stems (Figure 2C,D); and the leaves, reducing leaf area (Figure 4A) and biomass (Figure 4B). This ultimately impairs the correct in-vitro growth and the development of olive plants. In-vitro culture of plant tissues is usually carried out at a relative low PPF (37–70 μmol m^−2^s^−1^ PPF) [51] since it is assumed that photosynthesis is limited by the low CO_2_ concentration in the vessel, while this limitation is counteracted by sucrose supply in the medium [52].

The stimulation of stem elongation in response to a higher red/blue ratio has also been described in other micropropagated species. Conversely, blue light is known to inhibit stem elongation [15,33,53,54,55,56,57]. These responses are associated with differences in the relative contribution of blue-sensitive photoreceptors (cryptochromes and phototropins) and phytochromes [8]. Blue light has an important role in chlorophyll formation and chloroplast development [33,58,59]. At the molecular level, blue light upregulates genes involved in the synthesis of enzymes involved in chlorophyll biosynthesis, promoting chlorophyll accumulation [60]. However, we have observed a reduction of leaf chlorophyll content when the red/blue ratio was decreased from 9:1 to 7:3 (Figure 6A,C). There are also reports showing that monochromatic blue light reduces chlorophyll content in some species, while no effect was found in other ones [61,62]. In any case, there is consensus that a minimal blue light proportion threshold, quantified as 7%, is required to avoid dysfunctions associated to very high red/blue ratios [63]. Another component of light quality affecting plant morphological responses is the red/far-red ratio, which regulates stem elongation, branching, leaf expansion, and reproduction [15]. Since no far-red light has been used in the LED illumination regimes assayed in this work, we can conclude that regulation of adequate red/blue ratios is sufficient to optimize the shoot elongation rate and the internodal length of micropropagated olive plants.

Light quality/intensity can be used to regulate leaf growth in-vitro. We observed a positive correlation between light intensity and both leaf area and leaf fresh weight (Figure 4A,B). This phenomenon, previously observed in other micropropagated plants [64], is probably a consequence of an increased net leaf photosynthesis [65]. Interestingly, this correlation was not observed in other tissues like the callus (Figure 3). Other authors, such as [66], reported that callus production was not influenced by light quality in in-vitro propagated quince, although far-red plus blue light reduced callus growth. However, it was reported that red light induced greater callus formation in explants of poinsettia cultivars, while the blue light exerted the opposite effect [67].

The stomatal distribution and development are highly variable between species, depending mainly on lighting conditions and CO_2_ concentration. Our results show that stomatal frequency in micropropagated olive is induced by higher light intensity (Figure 5B) and/or a higher proportion of blue light (Figure 5D). Similar results were reported in several species [33,55,65,68,69,70]. Stomatal frequency impacts water loss through transpiration and water use efficiency [71]. Thus, lower light intensity resulting in lower stomatal frequency is expected to reduce water loss through leaf transpiration as previously observed [71], possibly improving plant ex-vitro acclimatization.

## 4. Materials and Methods

### 4.1. Plant Material

Plantlets of two wild varieties belonging to different subspecies of *Olea europaea* were obtained by in-vitro germination: *Olea europaea* subsp. *europaea* genotype AMK34 and *Olea europaea* subsp. *guanchica* genotype GUA7 (Appendix A; [72]). Uninodal explants (1 cm length) with one axillar bud were replicated. Apical explants were avoided in order to assure homogeneous responses. Explants were incubated in sterile 200 mL SIGMA jars (Merck V0633; Sigma-Aldrich, Darmstadt, Germany) containing 30 mL of olive culture medium [73], supplemented with zeatin (1 mgL^−1^), 20 gL^−1^ mannitol and 6 gL^−1^ agar. The pH of the medium was adjusted to 5.7 before autoclaving at 121 °C for 20 min. Six explants were incubated per jar under sterile conditions, avoiding direct contact of axillary buds with the agar culture medium. Jar lids were sealed with breathable sealing film (Parafilm, Pechiney Plastic Packaging, Menasha, WI, USA) and incubated in the growth chamber for 69 days. Growing conditions were 23 ± 2 °C and a 16 h light photoperiod, under the lighting conditions described in Table 1. Twelve explants were evaluated per treatment and genotype.

### 4.2. Illumination Treatments

Light treatments, which included seven types of LED quality/intensity combinations and one fluorescent light, are summarized in Table 1. Blue/Red illumination was obtained with intercalated red and blue 14.5 W/m LED strips. The blue LED has a peak emission at 455 nm and the red LED at 630 nm. Red/Blue treatment combinations included: 90%/10% with a photosynthetic photon flux (PPF) of 56 μmol m^−2^s^−1^ or µE (R9B1-56); 80%/20% (R8B2-30) with 30 μmol m^−2^s^−1^ PPF; 70%/30% with 34 μmol m^−2^s^−1^ PPF (R7B3-34); 70%/30% with 52 μmol m^−2^s^−1^ PPF (R7B3-52); and 60%/40% with 40 μmol m^−2^s^−1^ PPF (R6B4-40). Red/Blue/White illumination was obtained with intercalated red, blue and white 14.5 W/m LED strips: Red/Blue 70%/30% plus white LED with 84 μmol m^−2^s^−1^ PPF (R7B3W-84); and Red/Blue 60%/40% plus white LED with 98 μmol m^−2^s^−1^ PPF (R6B4W-98). When white LED lighting was supplemented to the R/B lighting, a 50% white vs. 50% R/B was used. The R/B ratio was also adjusted within the white LEDs lighting in order to equal it to that of the R/B lighting in the corresponding treatment (R7B3 in the R7B3W-84 treatment and R6B4 in the R6B4W-98 treatment). The fluorescent lighting consisted of daylight (6500 °K) tubular fluorescent lamps with 137 μmol m^−2^s^−1^ PPF (TFL-137). The resulting spectral patterns are given in Appendix A.

For a given number of LED chips, changes in the light quality (R/B ratio) inherently lead to changes in the light intensity (photon flux). To overcome the problem that two variables are assigned to a single treatment, different approaches have been addressed. On the one hand, in order to compare the effect of different light intensities without modifying light quality, we have compared plant responses to the following treatments: (i) R7B3-34 vs. R7B3-52 vs. R7B3-84; (ii) R6B4-40 vs. R6B4-98. The R7B3-34 vs. R7B3-52 comparison was possible by increasing the number of chips without modifying the R/B ratio. The additional treatment R7B3-84 was applied through supplementation with white LEDs. The same approach was used with the R6B4-98 treatment to make possible the comparison of the R6B4-98 vs. the R6B4-40 treatments. On the other hand, in order to compare the effect of light quality (different R/B ratios) without modifying the intensity (so that a single variable is assigned to a single treatment) we have compared the plant responses to the following treatments: (i) R9B1-56 vs. R7B3-52, given that 56 vs. 52 μmol/m^2^s^−1^ light intensities are similar enough not to involve significant differences in plant responses; (ii) R8B2-30 vs. R7B3-34, given that 30 vs. 34 μmol/m^2^s^−1^ light intensities are similar enough not to involve significant differences in plant responses; (iii) R6B4-98 vs. R7B3-84, given that 98 vs. 84 μmol/m^2^s^−1^ light intensities are similar enough not to involve significant differences in plant responses.

### 4.3. Determination of Growth and Developmental Parameters

For the quantification of vegetative development, the following parameters were quantified during the course of explants growth: plant height, stem length, number of shoots, number of nodes, and number of leaves. After 69 days of growth, plantlets were harvested and fresh weight (FW) was obtained. Dry weight (DW) and water content (WC) was calculated after drying the explants at 70 °C for 72 h. WC was determined in leaves obtained from six plants, using all leaves per plant according to the following equation:WC (%) = [(100 × (FW − DW)] (FW)^−1^(1)


After removing plantlets from the jars and before oven incubation, the leaves were excised and scanned in an Epson Stylus DX4000 multifunction printer (Seiko Epson Corp., Owa, Japan) to obtain total leaf area through pixel quantification with the Software ‘Medición de Hojas v1.0′ (Developed at the Department of Ecology, University of Seville, Spain; Taguas and Rivero, 1989).

### 4.4. Determination of Photosynthetic Pigments

Photosynthetic pigments were extracted from fully expanded leaves using 0.1 g of fresh plant material in 10 mL of 80% aqueous acetone (*n* = 12). After filtering, 1 mL of the suspension was mixed with 2 mL acetone, and chlorophyll-a (Chl a), chlorophyll-b (Chl b) and carotenoid (Cx + c) content (μg g^−1^ FW) was determined using a spectrophotometer (U-2001; Hitachi Ltd., Tokyo, Japan) measuring the absorbance at 663 nm, 645 nm and 470 nm, respectively, according to Arnon (1959) [74].
Chlorophyll A (Chl_a_) = 0.0127 · D_663_ − 0.00269 · D_645_(2)

Chlorophyll B (Chl_b_) = 0.0229 · D_645_ − 0.00468 · D_663_(3)

Chlorophyll (Chl) = Chl_a_ + Chl_b_(4)

Carotenoids = ((1000 · D_470_) − (1.82 · Chl_a_) − (85.02 · Chl_b_))/198(5)


### 4.5. Number and Size of Leaf Stomata and Trichomes

Ten samples from three fully expanded leaves, each leaf belonging to a different plant in a total of three plants per treatment, were used. Analysis of abaxial leaf cells was carried out in epidermal impressions, performed as described in [75]. Number and size of leaf stomata and trichomes was quantified on a Zeiss Axioskop microscope equipped with Nomarski optics, AxioCam MRc5, and the Zeiss AxioVision software (Freeware ‘Carl Zeiss AxioVision Rel.4.9.1.0′ available at the Zeiss Homepage http://www.zeiss.com/ (accessed on 20 September 2021), Carl Zeiss Microscopy GmbH, Jena, Germany) as described in [76].

### 4.6. Statistical Analyses

Unless otherwise specified, the data analysis was carried out from 12 single-plantlet replications within each treatment. The data from all experiments were subject to analysis of variance (ANOVA) and levels of significance are indicated in the figures by asterisks: * *p* ≤ 0.05; ** *p* ≤ 0.01; *** *p* ≤ 0.001. Non-significant (ns) differences were indicated when *p* > 0.05. Multiple comparisons of means were analyzed by Tukey’s HSD (honestly significant difference). A multiple range test was calculated using the Statistical Analysis System (STATGRAPHICS Centurion XVI software; http://www.statgraphics.com (accessed on 20 September 2021); StatPoint Technologies, Warrenton, VA, USA). In the graphics where the correlation between the photosynthetic photon flux and different growth/developmental parameters were plotted, the regression lines and the coefficient of determination (R^2^) were obtained using the corresponding function in Excel (Microsoft Office 2016 for Windows; Microsoft Corporation, Redmond, Washington DC, USA).

## 5. Conclusions

The aim of this study was to evaluate the effect of light quality and intensity on olive micropropagation, also comparing specific responses to LED vs. conventional fluorescent illumination. We concluded that two different subspecies of *Olea europaea* showed better growth and development under LED illumination in comparison to fluorescent lamps. Seventy percent red and thirty percent blue LEDs with a PPF of 34 μmol m^−2^s^−1^ of light irradiance was the optimal lighting treatment for olive micropropagation in terms of lower energy consumption, higher height and biomass, accumulation of photosynthetic pigments, optimal manipulation of in-vitro explants, lower leaf area and minimal stomatal density.

## Figures and Tables

**Figure 1 plants-10-02214-f001:**
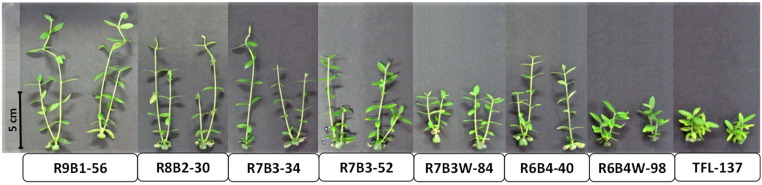
Effect of illumination quality and quantity on micropropagated olive plantlets. Different illumination regimes from light-emitting diodes (LEDs) and fluorescent lighting (TFL-137) were used for in-vitro clonal propagation of the AMK34 genotype of Olea europaea subsp. europaea. Shoot explants were cultivated for 69 days.

**Figure 2 plants-10-02214-f002:**
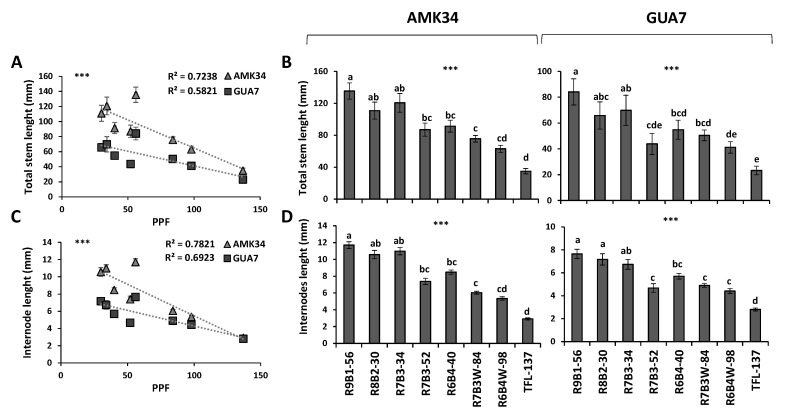
Effect of light quality and quantity on stem length and internodal elongation of micropropagated olive plantlets. Effect of different illumination regimes from light-emitting diodes (LEDs) and fluorescent lighting (TFL-137) on total stem length (**A**,**B**) and average internodal length (**C**,**D**) of different *Olea europaea* subspecies: subsp. *europaea* (genotype AMK34); and subsp. *guanchica* (genotype GUA7). PPF, photosynthetic photon flux. Level of significance: *** *p* ≤ 0.001.

**Figure 3 plants-10-02214-f003:**
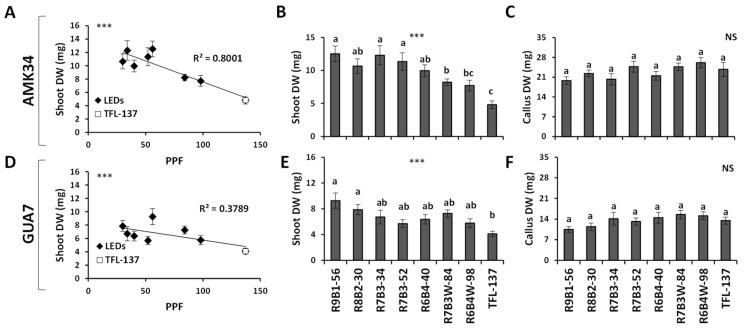
Effect of illumination quality and quantity on plant biomass. Correlation of light intensity with shoot dry weight (DW) in the olive genotypes AMK34 (**A**) and GUA7 (**D**). Effects of different illumination regimes from light-emitting diodes (LEDs) and fluorescent lighting (TFL-137) on shoot DW (**B**,**E**) and callus DW (**C**,**F**) of the olive genotypes AMK34 and GUA7, respectively. PPF, photosynthetic photon flux. Levels of significance: *** *p* ≤ 0.001; and non-significant (NS).

**Figure 4 plants-10-02214-f004:**
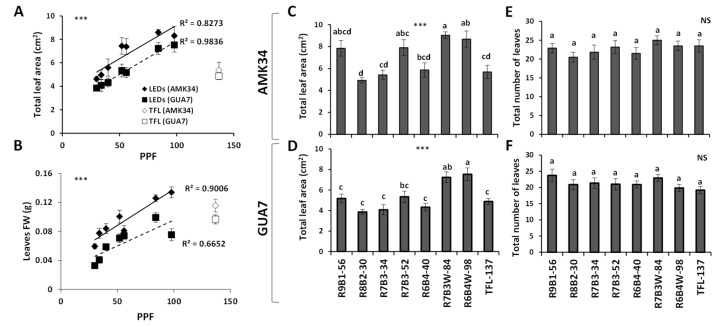
Effect of illumination quality and quantity on leaf area and leaf fresh weight. Correlation of light intensity with the leaf area in the AMK34 and GUA7 olive genotypes (**A**). Correlation of light intensity with the leaf fresh weight in the AMK34 and GUA7 olive genotypes (**B**). Effect of different illumination regimes from light-emitting diodes (LEDs) and fluorescent lighting (TFL-137) on the total leaf area of AMK34 (**C**) and GUA7 (**D**) plants. Effects of different illumination regimes from light-emitting diodes (LEDs) and fluorescent lighting (TFL-137) on the total number of leaves in AMK34 (**E**) and GUA7 (**F**) plants. PPF, photosynthetic photon flux. Levels of significance: *** *p* ≤ 0.001; and non-significant (NS).

**Figure 5 plants-10-02214-f005:**
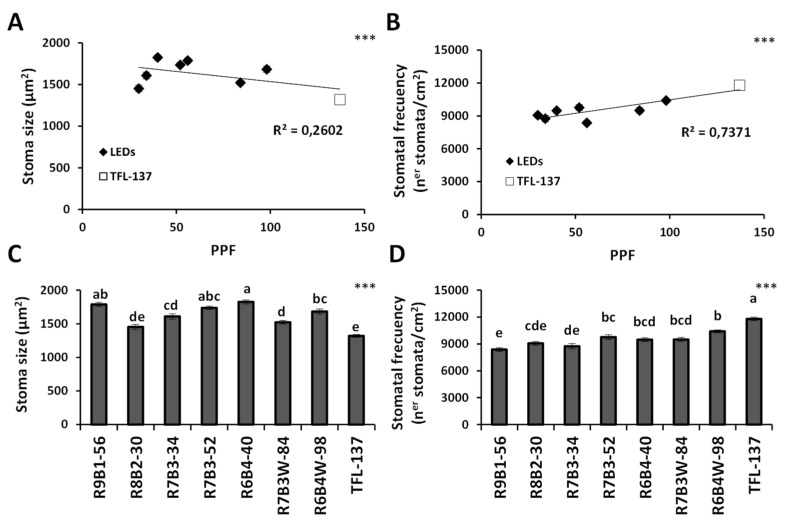
Effect of illumination quality and quantity on stomatal size and frequency. Correlation of light intensity with the stomatal size (**A**) and the stomatal frequency (**B**) in in-vitro grown AMK34 plants. Effect of different illumination regimes from light-emitting diodes (LEDs) and fluorescent lighting (TFL-137) on the stomatal size (**C**) and the stomatal frequency (**D**) in in-vitro grown AMK34 plants. PPF, photosynthetic photon flux. Levels of significance: *** *p* ≤ 0.001.

**Figure 6 plants-10-02214-f006:**
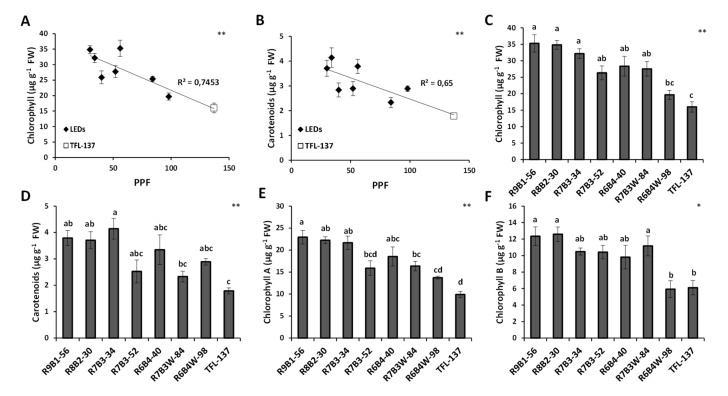
Effect of illumination quality and quantity on pigment composition. Correlation between light intensity and total chlorophylls (**A**) or carotenoid pigments (**B**) of micropropagated AMK34 plant leaves. Effects of different illumination regimes from light-emitting diodes (LEDs) and fluorescent lighting (TFL-137) on: total chlorophylls (**C**) and carotenoid pigments (**D**) in micropropagated shoots of AMK34 plants. Effects of different illumination regimes from light-emitting diodes (LEDs) and fluorescent lighting (TFL-137) on: chlorophyll-A (**E**) and chlorophyll-B (**F**). PPF, photosynthetic photon flux. Levels of significance: * *p* ≤ 0.05; ** *p* ≤ 0.01.

**Table 1 plants-10-02214-t001:** Lighting conditions assayed in this study.

Red/Blue Ratio (%)	Photon Flux (µmol m^−2^ s^−1^)	Nomenclature
LED 90/10	56	R9B1-56
LED 80/20	30	R8B2-30
LED 70/30	34	R7B3-34
LED 70/30	52	R7B3-52
LED 60/40	40	R6B4-40
LED 70/30 + white	84	R7B3W-84
LED 60/40 + white	98	R6B4W-98
Fluorescent	137	TFL-137

## Data Availability

Data supporting the findings of this study are available within the paper and within its Appendix A published online. Further information may be obtained from the corresponding authors.

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
