# Peer review of "Growth Quality and Development of Olive Plants Cultured In-Vitro under Different Illumination Regimes"

_plants, 2021, doi:10.3390/plants10102214_

Round 1
Reviewer 1 Report
Dear Authors,
The Authors took into account my suggestions from the previous review, that is why I recommend the article for publication
Author Response
Thank you very much
Reviewer 2 Report
I have completed the review of the article: ''Growth quality and development of olive plants cultured in vitro under different illumination regimes.''
The paper addresses a topical issue, namely the effect of light quality and intensity in the production of in vitro cultures, especially since, according to the author (L87), this type of study has not been done on Olea europaea species. In my opinion, the strengths of the article are: i) an experimental design in which several combinations of red and blue LEDs light are used whose intensity varies from 30 to 98 PPF compared to the light from the fluorescent tube with an intensity of 137 PPF; ii) and a statistical analysis based on correlation, ANOVA, and Tukey HSD of some morphological and physiological parameters. The results are interesting and surprising and can contribute to the development of the in vitro culture methodology, contributing to the increase of the energy efficiency and to the obtaining of better production results. The paper is written clearly and concisely, the results are new and meet the criteria to be published in Plants.
Specific comments
L130 Figure 2, I think you should specify in the description of the figure what is PPF. Comment valid for all figures containing PPF.
L137-155, please discuss the result obtained in Figure 3 A and D on the correlation between light intensity and dry biomass.
L 170 Fig 4 B please specify why the correlation was made only for AMK34. The same comment for Figures 5 and 6.
Please pay attention to the presentation of the results, especially those related to the correlations, because I think that the text can be improved in order to be easier to understand and to be in agreement with what we can see in the Figures.
L300 please use italics for species.
L384-392 please present the correlation method. Specify for which species it was made and justify why. I say this in accordance with the above, because in Figures 4, 5, and 6 only the correlations for AMK34 are shown.
Please include a paragraph for conclusions.
Author Response
REVIEWER 2
Specific comments:
L130 Figure 2, I think you should specify in the description of the figure what is PPF. Comment valid for all figures containing PPF.
Definition of PPF, photosynthetic photon flux, was included in the description of figures 2 to 6.
L137-155, please discuss the result obtained in Figure 3 A and D on the correlation between light intensity and dry biomass.
The results obtained in Figure 3 A and D have been more fully described in both the “Results” and, specially, the “Discussion” sections of the manuscript:
- L143-145. Thus, a negative correlation was observed between light intensity (PPF) and dry biomass of in-vitro grown plants. (Fig. 3A and 3D).
- L248-256. We show in this work reduction of shoot biomass (Fig. 3A-D) and lower pigments content (Fig. 6) in response to higher light intensity, indicating that PPF values higher than 50 μmol m-2s-1 impairs in-vitro culture of olive plants. Previous authors also observed a decrease of dry weight in plants grown in-vitro at high light intensity [49,50]. This phenomenon is probably the consequence of a suboptimal cell elongation rate under high light intensity, affecting different photosynthetic organs: the internodes, producing abnormally short stems (Fig. 2C-D); and the leaves, reducing leaf area (Fig. 4A) and biomass very significantly (Fig. 4B). This ultimately impairs the correct in-vitro growth and development of olive plants.
L 170 Fig 4 B please specify why the correlation was made only for AMK34. The same comment for Figures 5 and 6.
The correlation of PPF vs leaf biomass in GUA7 (R2= 0.6652) has been also included in Fig. 4B. We initially performed all the measurements with both wild olive genotypes (AMK34 and GUA7). The results were always very similar between the two genotypes: stem size (Fig. 1 and 2); shoot biomass (Fig. 3); and leaf area and biomass (Fig. 4). Due to the very identical behaviour between the two genotypes studied, measurements of stomatal size and density, as well as the chlorophyll and carotenoid content was only quantified in AMK34. All these measurements were repeated twice, with very similar results.
Please pay attention to the presentation of the results, especially those related to the correlations, because I think that the text can be improved in order to be easier to understand and to be in agreement with what we can see in the Figures.
Modifications have been made to the text to achieve a clearer, simpler and more direct message, especially those related to the correlations. For example, values of the linear regression lines have been removed from the text given that they are already present in the figures. See Lines 106-109; 143-145; 162-163
L300 please use italics for species.
Done!
L384-392 please present the correlation method. Specify for which species it was made and justify why. I say this in accordance with the above, because in Figures 4, 5, and 6 only the correlations for AMK34 are shown.
The correlation method has been included in the “Materials and Methods” section.
- L388-392: In the graphics where the correlation between the photosynthetic photon flux and different growth/developmental parameters were plotted, the regression lines and the coefficient of determination (R2) was obtained using the corresponding function in Excel (Microsoft Office 2016 for Windows; Microsoft Corporation, Redmond, Washington, USA). Parameters were measured and correlations were obtained for both wild olive genotypes in Figs 2, 3 and 4. Parameters were measured and correlations were obtained only for AMK34 in Figs 5 and 6 due to the reasons explained before.
Please include a paragraph for conclusions.
In the previous version of the manuscript, conclusions were given in the last paragraph of the discussion section. Now, it is present as section 5 (page 13) in the new version.

This manuscript is a resubmission of an earlier submission. The following is a list of the peer review reports and author responses from that submission.
Round 1
Reviewer 1 Report
As shown in Table 1, all light quality treatments have different light intensities except for R7B3. Two independent variables are assigned to a single treatment without block design; Since the variables are not properly controlled, it is not possible to derive any conclusion from the results. I understand the effort of authors put into this manuscript, but I have to say that this manuscript is not qualifying for a scientific paper.
other comments:
- In two of the "LED+white" treatments, what was the ratio between RB and white LEDs? it is not specified.
- If I understood correctly, the focus of this study is the influence of light regime on in vitro micro-propagated olive plantlets. But the introduction does not provide sufficient background information. (i.e. on in vitro micropropagation)
- the rationale of the study is too brief (L63-65); it needs to be expanded significantly.
- sentences are not logically connected and somewhat mixed up in the third paragraph (L44-60).
- need to describe methods more clearly.
L75 "200ml SIGMA jars" - c.f. "125-ml jars (NALGENE 2118-0004; Thermo Scientific, Hudson, NH, USA)"
L80 "breathable plastic film"
L85-98 textual description of spectra is not sufficient for the interpretation or reproduction of this study. spectral data are required.
L90 "µM" - the exact expression is "µmol". M is mol per liter and E (Einstein) is not a SI unit.
L106 "until they reached constant weight" - should be presented quantitatively. i.e. drying time in hours.
L119 "a Hitachi U-2001 spectrophotometer (Hitachi Ltd, Japan)" - "a spectrophotometer (U-2001; Hitachi Ltd, Tokyo, Japan)"
L127 "a Zeiss Axioskop microscope equipped with Nomarski optics"
Figure 1: baseline alignment of samples, clear separation between treatments, background separation (color), etc.
Reviewer 2 Report
Dear Authors,
The submitted article concerns the influence of different types and intensity of light on selected morphological and physiological processes of olive.
The abstract is succinctly written. Just check that the number of words is acceptable.
In the introduction, I propose to add a description of the species and its environmental requirements. This will allow you to visualize what plant is being talked about. Maybe the authors have photos of this genre, for the variety of work it would be worth adding juxtaposed photos of this genre. Please also specify the purpose of the paper. In this form we ca nread about many different things. Wouldn't it be better to write: The aim was to investigate selected morphological and physiological parameters of olive plants by measuring and list the examined features (parameters)... - in the appropriate order. My suggestion is for the sake of clarity in reading the text.
The material and methods should be included in the discussion, please check the template for the preparation of the article for publication.
Here please add the formulas for chlorophylls and carotenoids, same as for water content (WC).
This part describes the results obtained, not discussing them. Here, we do not cite the literature, we only describe the results. Please see the attachment.
The description of the results is acceptable. However, I note the significance - please change them to letters in all figures. Please delete the cited literature and comments. Here, there is only a list of the obtained results, similarities and differences.
The discussion is the part that concerns the discussion of own results in relation to the literature available in this topic. Here you should cite your results in the form of tables and figures (all) and discuss the topic briefly but thoroughly. Suggestions in the attachment.
Conclusions are a form of summarizing what was the aim of our work and drawing the conclusions. Please refer to the work's objectives and draw conclusions. In this form, the conclusion are not suitable.
In terms of editing, the paper must be redrafted; in this form it does not meet the journal's template.
